# Genome-Wide Identification of Common Bean *PvLTP* Family Genes and Expression Profiling Analysis in Response to Drought Stress

**DOI:** 10.3390/genes13122394

**Published:** 2022-12-16

**Authors:** Xue Dong, Huijun Zhu, Xiaopeng Hao, Yan Wang, Xiaolei Ma, Jiandong Zhao, Jianwu Chang

**Affiliations:** 1Center for Agricultural Genetic Resources Research, Shanxi Agricultural University, Taiyuan 030031, China; 2Key Laboratory of Crop Gene Resources and Germplasm Enhancement on Loess Plateau, Ministry of Agriculture, Taiyuan 030031, China; 3Shanxi Key Laboratory of Genetic Resources and Genetic Improvement of Minor Crops, Taiyuan 030031, China; 4National Laboratory of Minor Crops Germplasm Innovation and Molecular Breeding, Shanxi Agricultural University, Taiyuan 030031, China; 5College of Agriculture, Shanxi Agricultural University, Taiyuan 030031, China

**Keywords:** common bean, lipid transfer protein, tandem duplicates, drought stress, expression profile

## Abstract

Common bean is one of the most important legume crops for human consumption. Its yield is adversely affected by environmental stress. Plant non-specific lipid transfer proteins (nsLTPs) are essential for plant growth, development, and resistance to abiotic stress, such as salt, drought, and alkali. However, changes in *nsLTP* family genes responding to drought stress are less known. The *PvLTP* gene family in the common bean was identified by a comprehensive genome-wide analysis. Molecular weights, theoretical isoelectric points, phylogenetic tree, conserved motifs, gene structures, gene duplications, chromosome localization, and expression profiles were analyzed by SignalP 5.0, ExPASy, ClustalX 2.1, MEGA 7.0, NCBI-CDD, MEME, Weblogo, and TBtools 1.09876, respectively. Heatmap and qRT-PCR analyses were performed to validate the expression profiles of *PvLTP* genes in different organs. In addition, the expression patterns of nine *PvLTP* genes in common beans treated with drought stress were investigated by qRT-PCR. We obtained 58 putative *PvLTP* genes in the common bean genome via genome-wide analyses. Based on the diversity of the eight-cysteine motif (ECM), these genes were categorized into five types (I, II, IV, V, and VIII). The signal peptides of the PvLTP precursors were predicted to be from 16 to 42 amino acid residues. PvLTPs had a predicated theoretical isoelectric point of 3.94–10.34 and a molecular weight of 7.15–12.17 kDa. The phylogenetic analysis showed that *PvLTPs* were closer to *AtLTPs* than *OsLTPs*. Conserved motif and gene structure analyses indicated that *PvLTPs* were randomly distributed on all chromosomes except chromosome 9. In addition, 23 tandem duplicates of *PvLTP* genes were arranged in 10 gene clusters on chromosomes 1 and 2. The heatmap and qRT-PCR showed that *PvLTP* expression significantly varied in different tissues. Moreover, 9 *PvLTP* genes were up-regulated under drought treatment. Our results reveal that *PvLTPs* play potentially vital roles in plants and provide a comprehensive reference for studies on *PvLTP* genes and a theoretical basis for further analysis of regulatory mechanisms influencing drought tolerance in the common bean.

## 1. Introduction

The common bean (*Phaseolus vulgaris* L.) is an important legume used as food because it is a major source of proteins, minerals, and vitamins [1]. Thus, it is consumed as part of the traditional diets in Europe (e.g., the Mediterranean region) and the Middle East [2]. The increase in living standards and growing demand for healthy food have necessitated increases in common bean yields. Because of its significant economic and production value, the common bean has become one of the most profitable cultivated legumes in the worldwide market over the last two decades. However, its growth and yield are restricted by biotic and abiotic stresses, such as anthracnose, fusarium wilt, drought, and salt [3]. The yield of common bean decreased the most among all legumes under drought stress [2,4,5]. According to Daryanto et al., the common bean needs to be improved for drought resistance due to its significance in world production and human nutrition [4]. A few studies identified the functions and mechanisms of drought resistance-related genes in the common bean [6,7,8]. The transcription of *PvXIP1;2*, which encoded a drought-related aquaporin, was strongly induced by drought stress. Moreover, the overexpression of *PvXIP1;2* in *Arabidopsis* increased the survival rate and proline content but decreased ion leakage and the malondialdehyde content. Furthermore, compared with control roots, *PvXIP1;2*-overexpressing common bean hairy roots reportedly had significantly higher water contents and growth rates [6]. The expression of another aquaporin gene (*PvTIP1;1*) was quickly down-regulated in drought-stressed leaves but returned to normal or higher levels after rehydration, suggesting *PvTIP1;1* might be associated with high water permeability. Hence, the down-regulated expression of this gene may help prevent water loss, thereby protecting plants from the detrimental effects of drought [7,8]. Therefore, there is an urgent need to screen for drought-resistance genes, which upon modification, could improve the common bean’s resistance to drought stress by adjusting its growth, metabolism, and cellular structures.

Plant non-specific lipid transfer proteins (nsLTPs) are small molecular proteins that transport hydrophobic compounds, including acyl-coenzyme A, glycolipids, and fatty acids [9,10,11,12]. The N-terminus of nsLTP contains a signal peptide sequence, which guides nsLTP proteins that are being secreted to the cytoplasm [13]. Eight-cysteine motifs (ECM, C-Xn-C-Xn-CC-Xn-CXC-Xn-C-Xn-C) in mature nsLTP proteins form four internal disulfide bonds, producing a stable hydrophobic tertiary structure [14]. This structure can bind different hydrophobic compounds and lipid molecules in cells [15,16].

Plant nsLTP belongs to a multi-gene family and is classified into two types: LTP1 and LTP2 [9]. LTP1 has a molecular weight of about 9 kDa, 90–95 amino acid residues in mature proteins, and a signal peptide with 21–27 amino acid residues. The molecular weight of LTP2 is about 7 kDa, with approximately 70 and 27–35 amino acid residues for mature proteins and the signal peptide, respectively [17,18]. Increasing nsLTPs have been identified, thus rendering their traditional classification methods insufficient. Boutrot et al. reclassified nsLTP into nine types (I–IX) based on the ECM interval characteristics and amino acid sequence similarity [19]. The X and XI types were also added recently [20,21]. Following the development of genome sequencing technology, the *nsLTP* gene family has been characterized in various plants. For instance, *Arabidopsis*, rice, and *Nicotiana tabacum* have 45 members in eight types (I–VI, VIII, and IX), 49 in eight types (I–VIII), and 100 in six types (I, II, IV, V, VII, and VIII), respectively [19,22].

Studies showed that nsLTPs participated in various biochemical and physiological pathways of plant growth and development, including signal transduction, keratin synthesis, anther development, and seed maturation [23,24,25]. Additionally, nsLTPs also played regulatory roles in plant response to abiotic stresses and defense against pathogens. Some researchers attributed nsLTPs to plant pathogenesis-related proteins of the PR-14 family [26,27,28,29]. Recently, researchers have focused on plant resistance to abiotic stresses. Overexpression of *AZI1*, a member of the *nsLTP* family in *Arabidopsis*, resulted in a salt-tolerant phenotype. Further research showed that AZI1 combined with mitogen-activated protein kinase 3 (MPK3) forms a complex, which MPK3 positively regulates to resist salt stress [30]. Rice *OsDIL1* and foxtail millet *SiLTP* were induced by drought, salt, and abscisic acid (ABA) [31,32]. Overexpressing *SiLTP* could enhance drought and high-salt stress resistance, and the RNA interference of *SiLTP* led to sensitive phenotypes [32]. In drought-tolerant wheat lines, the expression of the *TdLTP4* gene was significantly higher than that of drought-sensitive lines. The heterologous expression of *TdLTP4* in *Arabidopsis* led to tolerance to abiotic and biotic treatments [33]. Because of their association with stress resistance-related transcription factors, researchers believe that *nsLTP* can enhance stress resistance in plants.

Until now, only four *nsLTP* genes have been found in the common bean. Among these, the expression of *PvLTP24* increased significantly in stems and leaves following drought and ABA treatments [34]. The apical cortex specifically expressed *PVR3* in the root tip, which could be used as a marker gene for cortex development [35,36]. Additionally, *PvLTP1a* and *PvLTP1b* are the two major food allergens, while the functions of the other common bean *nsLTP* genes are still unclear [37]. We utilized bioinformatics methods to identify the common bean’s *PvLTP* gene family. Chemical and physical properties, as well as a phylogenetic tree, conserved motifs, gene structure, and chromosome location, were analyzed. Moreover, a qRT-PCR assay was used to analyze the expression levels of different *PvLTP*s in organs and drought stress. Our findings will serve as a foundation for gene function and mechanism studies of the common bean *PvLTP*s in response to drought stress.

## 2. Materials and Methods

### 2.1. Identification and Bioinformatic Analyses of PvLTP Family in Common Bean

The genome and proteome sequences of common bean, *Oryza sativa,* and *Arabidopsis thaliana* were downloaded from Phytozome (http://phytozome.jgi.doe.gov/, accessed on 1 March 2021), RAP (https://rapdb.dna.affrc.go.jp/, accessed on 1 March 2021), and TAIR (https://www.arabidopsis.org/, accessed on 1 March 2021) databases, respectively [38,39,40]. The *NsLTP* sequence ID lists of *Arabidopsis* and rice were obtained from Boutrot et al. [19]. Candidate *nsLTP* genes of the common bean were obtained using BLASTp with AtLTP and OsLTP sequences as queries to blast against the common bean protein database with an E-value of 1 × 10^−5^. Proteins with the HMM domain PF00234 were further retrieved using HMMER 3.3 with default parameters to avoid missing *nsLTP* genes. After eliminating redundant sequences, all proteins of putative nsLTP genes lacking the essential ECM domain were removed manually. Subsequently, the signal peptide cleavage sites were analyzed using Signal 5.0 (https://services.healthtech.dtu.dk/service.php?SignalP-5.0, accessed on 5 March 2021). Proteins without N-terminal signal sequences and the deduced hybrid proline-rich proteins were discarded. Moreover, the rest of the candidate sequences were subjected to BLASTp to exclude potential α-amylase inhibitors and cereal storage proteins using RATI and 2S-albumin as queries, respectively [41,42]. The remaining predicted proteins were uploaded to the NCBI-CDD (https://www.ncbi.nlm.nih.gov/cdd, accessed on 7 March 2021) and Pfam (http://pfam.xfam.org/, accessed on 7 March 2021) websites to identify the conserved LTP domain. Finally, the mature proteins containing more than 120 amino acid residues were eliminated. The identified *PvLTPs* were named according to Boutrot’s method and their orders on chromosomes [19].

### 2.2. Sequence Alignment and Phylogenetic Analysis

The ECM domains of PvLTPs from *Arabidopsis*, rice, and the common bean were subjected to multiple alignments by ClustalX 2.1 software with default parameters [43]. After that, they were utilized to build a phylogenetic tree based on the Neighbor-Joining model with MEGA 7.0 software [44,45]. The phylogenetic tree was visualized with Figtree 1.4.4 (http://tree.bio.ed.ac.uk/software/Figtree/, accessed on 8 March 2021). For statistical reliability, bootstrap tests were computed with 1000 replications.

### 2.3. Conserved Motifs and Gene Structure Analysis

To identify conserved motifs, PvLTPs were uploaded to the MEME (https://meme-suite.org/meme/tools/meme, accessed on 10 March 2021) website with the motif width setting as 6–50 amino acid residues and the maximum motif number setting as 10 [46]. Gene structures were analyzed and visualized using TBtools 1.09876 [47]. The WebLogo tool (http://weblogo.threeplusone.com/, accessed on 10 March 2021) was performed to draw sequence logos of the conserved ECM domain [48].

### 2.4. Chromosomal Localization and Gene Duplication

The common bean v2.1 gff3 file was retrieved from the Phytozome website [38,49]. The relative distances and positions of *PvLTPs* were obtained according to the genome annotation information and drafted on the 11 chromosomes using TBtools 1.09876 [47]. The duplication events of *PvLTP* gene family members in the common bean were identified using TBtools 1.09876 with the default parameters.

### 2.5. Plant Materials

Common bean cultivars (Pinjinyun 3) were planted in the field at the Center for Agricultural Genetic Resources Research at Shanxi Agricultural University (China, Shanxi, Latitude 112°58′ E, 37°78′ N, Appendix A). The roots, stems, leaves, flowers, seeds, and pods were collected and fast-frozen in liquid nitrogen before being preserved at −80 °C. Moreover, the cultivar seeds were cultivated at 25 °C with 14 h of light and 20 °C with 10 h of darkness in a growth chamber (Appendix A). The seedlings were treated with 20% PEG6000 for 24 h at the trifoliate leaves stage. The leaves were collected after 0, 6, 12, and 24 h under drought treatment, and then samples were stored at −80 °C before RNA extraction. For each organ, three separate biological replicates and technical replicates were performed.

### 2.6. Quantitative Real-Time PCR Analysis

Total RNA was extracted from roots, stems, flowers, leaves, seeds, and pods using SV Total RNA Isolation System (Promega, Madison, WI, USA) as directed by the manufacturer’s instruction. Precisely 1 μg of total RNA was reverse-transcribed by PrimeScript RT Master Mix (TaKaRa, Otsu, Japan). Then, the cDNA was amplified on a Quantstudio 6 thermal cycler (Applied Biosystems, Waltham, MA, USA) using TB Green Premix Ex Taq II (TaKaRa, Otsu, Japan). The qRT-PCR amplification procedure was 95 °C for 30 s, performed with 40 cycles of 95 °C for 5 s and 60 °C for 34 s. We used the common bean *Actin* gene as a reference control. For each experiment, three separate technical and biological replicates were performed. The 2^−ΔΔCt^ analysis method was used to calculate the relative transcription levels (Appendix A) [50]. All primers are listed in Appendix A.

## 3. Results

### 3.1. Genome-Wide Identification of Putative nsLTPs in Common Bean

To investigate all putative *nsLTP*s in the common bean, we performed BLASTp using AtLTP and OsLTP sequences as queries to search against the common bean proteome database. A total of 73 candidate *nsLTP* genes were obtained. Another ten *nsLTP* genes were retrieved from the HMM domain profiles of PF00234 and PF14368. After integrating the above results, three proteins lacking the essential ECM domain were manually removed. In addition, six proteins without N-terminal signal sequences were identified using Signal 5.0 and were deleted. Five hybrid proline-rich proteins were not taken into consideration. Proteins with high similarity to α-amylase inhibitors or cereal storage proteins were not found. The remaining 69 candidate protein sequences were uploaded to the Pfam and NCBI-CDD websites to identify the conserved LTP domains. All candidates possessed the LTP domains. Because all mature nsLTPs have low molecular weights, eleven of the sixty-nine candidates with more than 120 amino acid residues were discarded. As a result, 58 genes coding nsLTPs were designated as *PvLTPs*.

### 3.2. Classification and Sequence Analysis of PvLTPs

*PvLTPs* were classified using the sequence similarity method. The results indicated that the 58 *PvLTP* genes were categorized into five types, namely types I (*PvLTPI.1-45*), II (*PvLTPII.1*-*5*), IV (*PvLTPIV.1*-*5*), V (*PvLTPV.1*-*2*), and VIII (*PvLTPVIII.1*) (Table 1). No *PvLTP* gene was categorized into types III, VI, or VII. Obviously, the ECM domains were highly conservative in structure, except for the number of variable inter-cysteine residues (Table 2). Similar to rice and potato, the common bean genome contained the largest proportion of type I *PvLTP* genes (77.59%, 45 out of 58).

The characteristics of *PvLTP* genes and proteins, such as coding sequence (CDS) length, signal peptide and protein sequences, were analyzed using biological software. The length of CDS ranged from 246 (*PvLTPII.2*) to 465 bp (*PvLTPVIII.1*). All PvLTP precursors had a signal peptide varying in length from 16 (PvLTPI.8-10, PvLTPI.14) to 42 (PvLTPVIII.1) amino acid residues. Mature PvLTPs were usually small and had a low molecular weight ranging from 7.15 (PvLTPII.5) to 12.17 kDa (PvLTPVIII.1). The predicted theoretical isoelectric point varied from 3.94 (*PvLTPI.45*) to 10.34 (*PvLTPI.45*). The mean molecular weight was 9.82 kDa, and the isoelectric point was 7.52 (Table 1).

### 3.3. Phylogenetic Analysis of PvLTPs

To demonstrate the evolutionary relationship of *nsLTP* genes among the common bean, *Arabidopsis*, and rice, phylogenetic analysis was completed using ClustalX 2.1 and MAGA 7.0 software. As shown in Figure 1, the 152 *nsLTP* genes were divided into nine distinct clades, with 77, 33, 5, 14, 9, 8, 1, 3, and 2 *nsLTP* genes from types I–IX in each clade, respectively. Obviously, the largest portions of common bean *PvLTPs* and *Arabidopsis AtLTPs* were clustered together in clade I, whereas rice *OsLTPs* were mainly gathered in clade II. In addition, *PvLTPs* were closer to *AtLTPs* than *OsLTPs* in each clade of the phylogenetic tree. The cluster separation revealed a closer genetic lineage-specific relationship between the common bean and *Arabidopsis* during dicotyledon evolution.

### 3.4. Conserved Motifs and Structure of PvLTPs

CDD and Weblogo websites showed the conserved domains in the *PvLTP* gene family (Figure 2). PvLTP family members contained an ECM domain, which belonged to the α-amylase inhibitors and lipid transfer/seed storage proteins (AAI-LTSS) superfamily. A total of ten distinct conserved motifs (motif1–motif10) were identified using MEME tools (Figure 3) to analyze the relationship between the structural features and diversity of PvLTP members further. All PvLTP family members contained motif3 with highly conserved amino acid sequences. Moreover, motif1 and motif4 were presented in type I PvLTPs (except PvLTPI.6), while motif7 and motif10 were presented in type II PvLTPs. The results revealed that motif3 was conserved within the five PvLTP types.

The amount and distribution of exons and introns in *PvLTPs* were examined to investigate the components of *PvLTP* gene structure. Among the 58 *PvLTP* genes, 29 contained one intron and two exons, including 25 type I, one type IV, two type V, and one type VIII *PvLTPs*. The other 29 had no introns. Interestingly, all type II *PvLTP* genes had no introns, consistent with the findings in cotton and potato. These findings indicated that the *PvLTP* gene family had been conserved in plant evolution.

### 3.5. Chromosomal Localization and Gene Duplication of PvLTPs

To better obtain the distribution and exact position of *PvLTP*s on chromosomes, a detailed chromosome map was constructed. The 58 *PvLTP* genes were unevenly dispersed on ten chromosomes (Figure 4). Among them, twenty, seventeen, five, four, four, three, one, one, and one PvLTPs were located on chromosomes 8, 6, 10, 1, 7, 4, 2, 3, and 11, respectively, while there was no PvLTP on chromosome 9.

Gene duplication analysis showed that, among the 58 *PvLTPs*, 23 genes were arranged in ten gene tandem duplicate clusters and distributed on two chromosomes. Among the ten gene clusters, nine clusters contained 21 genes and were located on chromosome 1, and the other cluster was located on chromosome 2. The large proportion (39.66%) of tandem duplicate clusters indicated that tandem duplications might cause *PvLTP* gene family expansion in the common bean genome.

### 3.6. Expression Pattern of PvLTPs in Different Organs

The expression profile of *PvLTP*s in different organs was visualized in a heatmap using RNA-seq data from Phyzotome to investigate the functions of *PvLTP*s during common bean growth. Figure 5 showed that 19 *PvLTPI* genes were highly expressed in root nodules but lower in other tissues. Moreover, the rest of the *PvLTP* genes showed a diverse expression profile in different tissues.

To further validate the above results, the expression of nine *PvLTP*s in roots, stems, leaves, flowers, seeds, and pods was detected using qRT-PCR (Figure 6). *PvLTPI.27*, *PvLTPI.28*, and *PvLTPI.42* were highly expressed in flowers, while *PvLTPI.44* and *PvLTPV.1* were highly expressed in seeds, consistent with the results shown in the heatmap. In addition, *PvLTPII.3* and *PvLTPV.2* were predominantly expressed in stems, while *PvLTPI.45* was strongly expressed in pods. These findings suggested that the differentially expressed *PvLTP*s might play various biological functions in the growth and development of the common bean.

### 3.7. Expression Analysis of PvLTPs under Drought Stress

Abiotic environmental conditions have direct impacts on plant growth. Therefore, it is significant to analyze how relatively resistant genes change under different stresses. The expression levels of 9 *PvLTP*s after drought treatment were examined using qRT-PCR to explore the functions of *PvLTPs* involved in drought stress. The results showed that all *PvLTPs* were up-regulated after drought stress (Figure 7). The relative expression levels of *PvLTPI.28*, *PvLTPI.41*, *PvLTPI.42*, *PvLTPI.44*, *PvLTPI.45*, and *PvLTPII.3* showed a similar trend of first increasing and then decreasing, reaching the maximum at 12 h. *PvLTPI.44* especially exhibited more than 115-fold up-regulation. Moreover, the relative expression levels of the other three *PvLTP*s, *PvLTPI.27*, *PvLTPV.1*, and *PvLTPV.2* showed a trend of gradual increase and reached the highest of 37-, 78-, and 142-fold up-regulation at 24 h compared with the control, respectively. These findings displayed that *PvLTPs* might function as important positive regulators influencing plant tolerance towards drought stress.

## 4. Discussion

Plants have different types and numbers of the *nsLTP* gene family. *Brassica rapa* has nine types (I–VI, VIII, IX, and XI) with 63 members, while Six *Solanaceae* species have five types (I, II, IV, IX, and X) with 135 members [20,21]. In this work, whole-genome identification of the common bean obtained 58 *PvLTP* family members unevenly distributed across ten chromosomes, except for chromosome 9. We found that these *PvLTP* family members could be categorized into five types (I, II, IV, V, and VIII). Nevertheless, the absence of some types indicated deletion during the evolution of the common bean. The phylogenetic tree results showed that the common bean *PvLTPs* were closer to *Arabidopsis AtLTPs* than rice *OsLTPs*. The relationship distance between the dicotyledons and monocotyledons in the plants’ classification was consistent with the previous studies [51], indicating that the *nsLTP* genes existed before the separation of monocotyledonous plants and were conserved during evolution. The internal structures of types I and II had similarities, suggesting that the same category of genes may have similar biological functions.

Studying gene expression patterns can provide a theoretical basis for gene functions. Results have shown that the *nsLTP* gene responded to drought and other abiotic stresses. In *Arabidopsis*, *LTP3* was highly expressed after 6 h of drought treatment. *AtLTP3* overexpression significantly enhanced tolerance to drought stress. Further studies showed that the upstream transcription factor, MYB96, regulated the expression of *AtLTP3* by binding to the *AtLTP3* promoter, thus enabling *AtLTP3* to participate in the drought resistance signaling pathway [26]. A comparison between AtLTP3 and PvLTPI.45 revealed that their amino acid sequences are approximately 41.53% similar. Additionally, *AtLTP3* was expressed at high levels in the leaves, flowers, and siliques, with peak levels detected in the siliques. The *PvLTPI.45* expression pattern was similar to that of *AtLTP3*, with the highest expression level detected in the pods, followed by the flowers and leaves. Under drought conditions, the *AtLTP3* and *PvLTPI.45* expression levels increased by about 4-fold and then decreased, suggesting they may have similar functions in plant responses to drought stress. Several findings indicated that the expression levels of *nsLTP* genes, including *OsDIL* (rice), *ScLTP* (sugarcane), and *NtLTP4* (tobacco), were up-regulated or down-regulated to varying degrees after drought, salt, and other abiotic stress treatments [31,52,53]. For example, *StnsLTP1* expression in potatoes increases in response to excessive heat, salt, and drought. The overexpression of *StnsLTP1* in potatoes led to enhanced tolerance to abiotic stresses because it activated antioxidative defense mechanisms and up-regulated the expression of stress-related genes [54]. These observations imply that nsLTP proteins are involved in multiple stress-induced signaling pathways. In this research, the expression levels of *PvLTPs* were increased to varying degrees after the drought stress treatment. For instance, *PvLTPI.44* and *PvLTPV.2* increased 115 and 142 times, respectively, compared with the control. Therefore, it can be speculated that *PvLTPs* play a vital function in drought stress response.

Gene tandem duplication during genomic DNA replication and recombination is the key driver for gene family amplifications [55,56,57]. In *Arabidopsis* and rice genomes, 15–20% of genes were composed of tandem repeats of gene clusters considered crucial for evolution, plant disease resistance, and abiotic stress responses [58]. Several tandem repeats of genes were found in the *nsLTP* gene family of angiosperms, including 47.82% (66/138) in cotton, 51.43% (36/70) in barley, and 53.01% (44/83) in potato [59,60,61]. This indicated that tandem repeats played essential roles in gene amplification during the evolution of the *nsLTP* gene family. In this work, chromosome mapping and gene structure revealed that the gene replication events occurred during genome amplification and evolution of the common bean. Twenty-three genes of the *PvLTP* gene family were categorized into 10 tandem repeats of gene clusters on chromosomes 1 and 2, accounting for 39.66% of the family members. Most of the genes exhibiting the same tandem repeats had a high similarity, suggesting that they might have functional similarities. For instance, the nucleic acid sequence similarity of *PvLTPI.12*/*PvLTPI.13*/*PvLTPI.14*/*PvLTPI.15* was 98.62%, and the protein similarity was 96.71%. However, the nucleic acid sequence similarity of *PvLTPI.41*/*PvLTPI.42* was only 52.18%, with a protein similarity of 58.47%. This implied that, although the *PvLTP* family was derived from the same ancestor, it changed when plants adapted to the external environment during evolution. Moreover, we found that *PvLTPI.42* was more highly expressed in flowers than in other tissues, while the *PvLTPI.41* expression was higher in the tissues except for roots. The different expression patterns suggest that the evolution of duplicate genes might lead to the emergence of functional diversity.

## 5. Conclusions

In summary, whole-genome identification of the common bean identified 58 members of the *PvLTP* gene family, which were divided into five types (I, II, IV, V, and VIII). Each member contained the conserved ECM domain. *PvLTP* genes were randomly distributed on ten chromosomes, except for chromosome 9, of which 23 members formed ten sets of tandemly repeating gene clusters. The organ-specific expression profiles of nine *PvLTP* genes differed significantly. In addition, exposure to drought stress up-regulated the expression of these genes. These findings provide important insights into the *PvLTP* genes in the common bean and form the theoretical basis of future research on *PvLTP* functions related to plant development and stress responses. Furthermore, our study will contribute to investigating the mechanisms underlying the effects of *nsLTPs* on plant responses to abiotic stresses.

## Figures and Tables

**Figure 1 genes-13-02394-f001:**
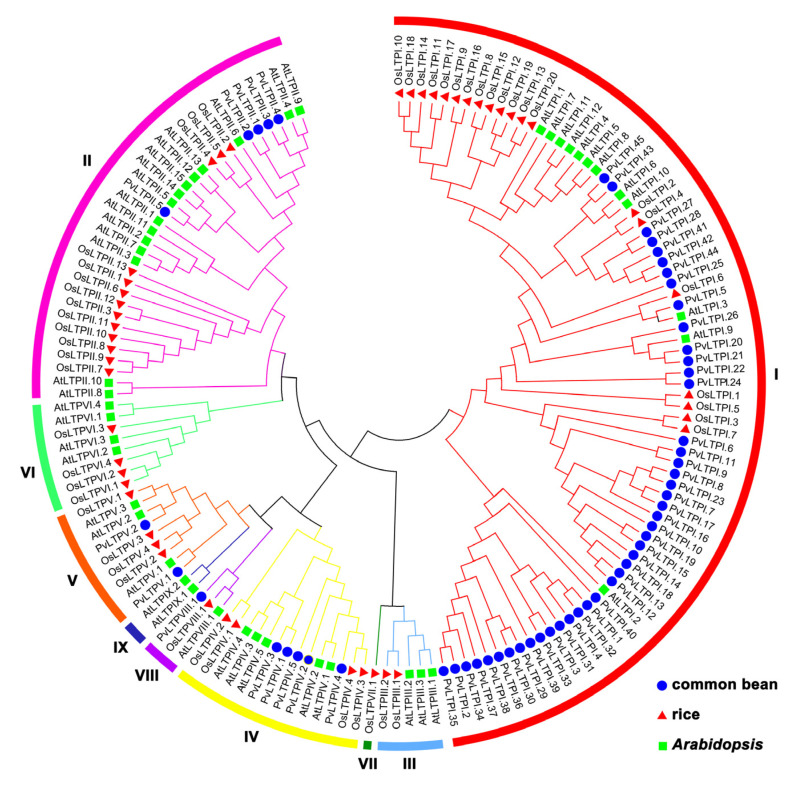
Phylogenetic tree of nsLTP proteins from the common bean, rice, and *Arabidopsis*. Nine nsLTP types are marked using different colors. Roman numbers indicate corresponding types of nsLTP family.

**Figure 2 genes-13-02394-f002:**
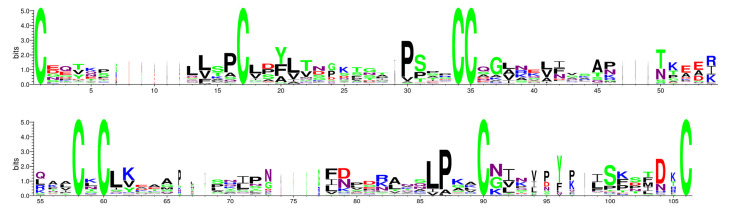
Conserved domain analysis of PvLTPs using the WebLogo website. The degree of conservation is indicated by the height of each amino acid. The number on the *x*-axis indicates its position in the ECM. The information measured is represented in bits on the *y*-axis.

**Figure 3 genes-13-02394-f003:**
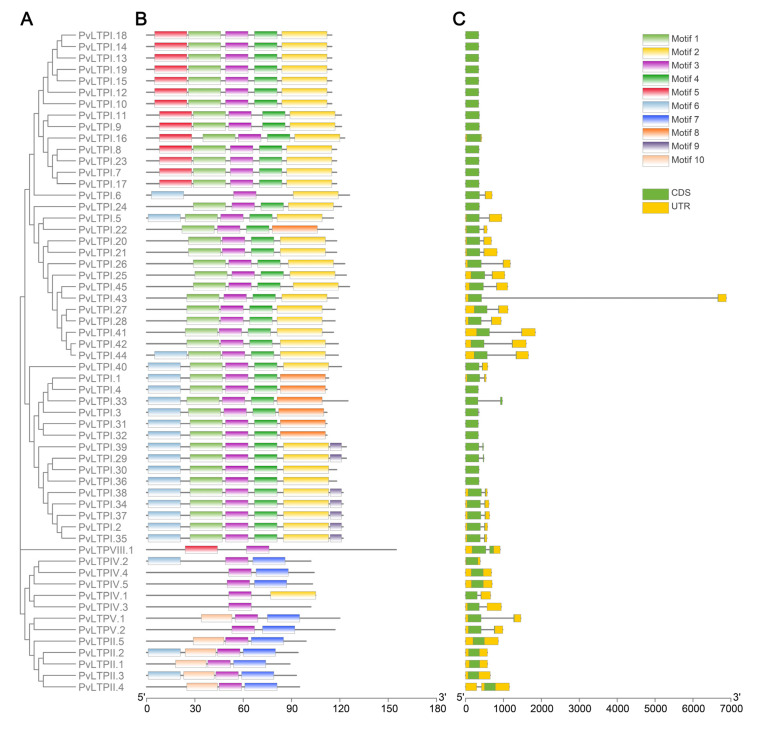
Phylogenetic tree, conserved motifs and gene structures of *PvLTP*s. (**A**) A Neighbor-Joining phylogenetic tree was generated with 1000 bootstrap replicates based on ECM domains of fifty-eight PvLTP proteins. (**B**) The conserved motifs are shown. A total of ten distinct conserved motifs were depicted using various colored boxes. (**C**) The gene structures of the *PvLTPs* were visualized. The yellow boxes, green boxes, and black lines indicate UTRs, exons, and introns, respectively.

**Figure 4 genes-13-02394-f004:**
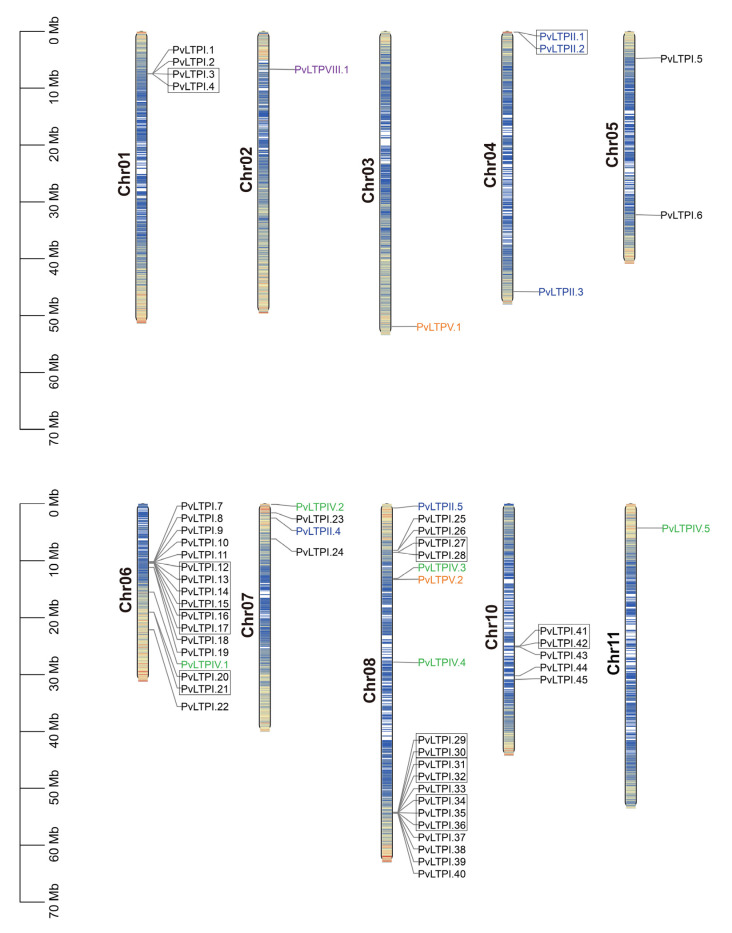
Chromosomal localization and tandem-duplicate genes of *PvLTP* family from the common bean. *PvLTP* types are marked with different colors. The tandem-duplicate genes are depicted with a black rectangle. Chromosomal distances are given in megabase (Mb).

**Figure 5 genes-13-02394-f005:**
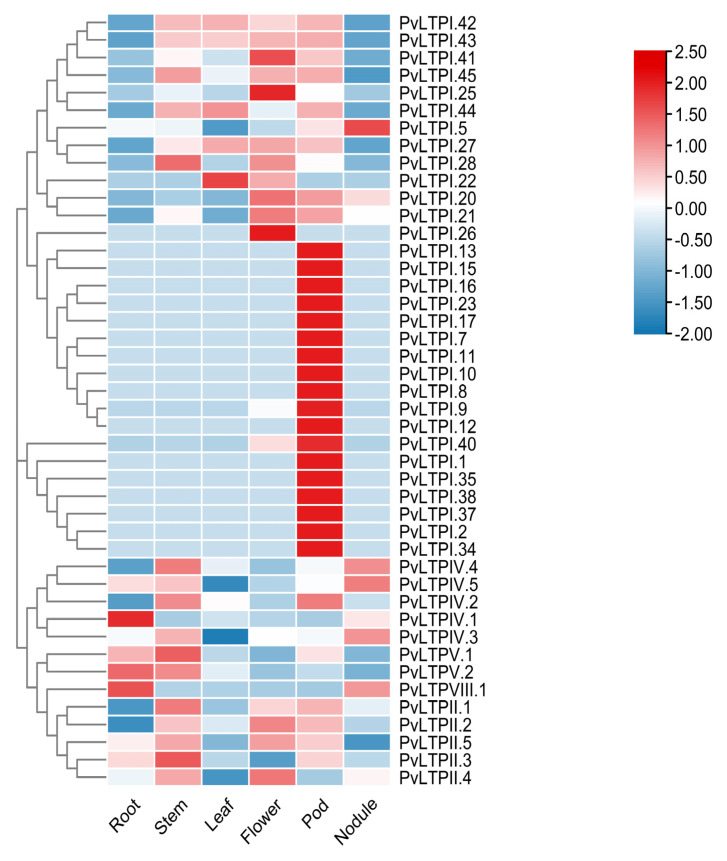
Heatmap of the *PvLTP* genes expression profiles in different organs. Blue, light blue, colorless, light red and red are used to show the differentially expressed levels. FPKM is used as the gene expression value unit.

**Figure 6 genes-13-02394-f006:**
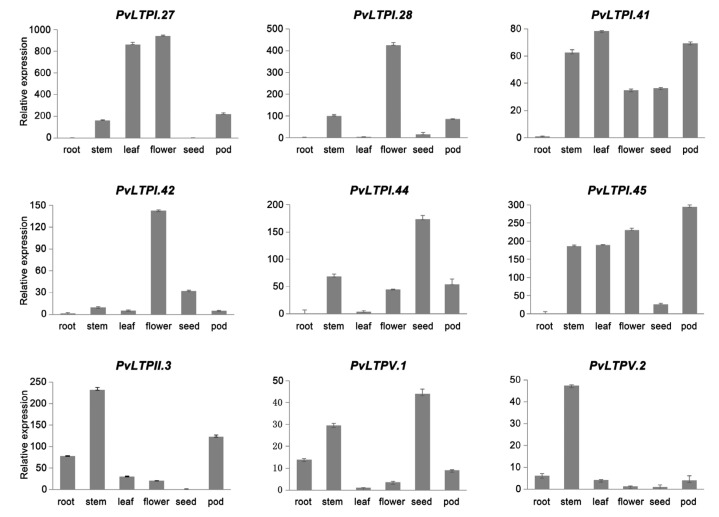
Expression levels of *PvLTPs* in six tissues by qRT-PCR. The internal reference gene used was *Actin*. Error bars reflect the standard deviation..

**Figure 7 genes-13-02394-f007:**
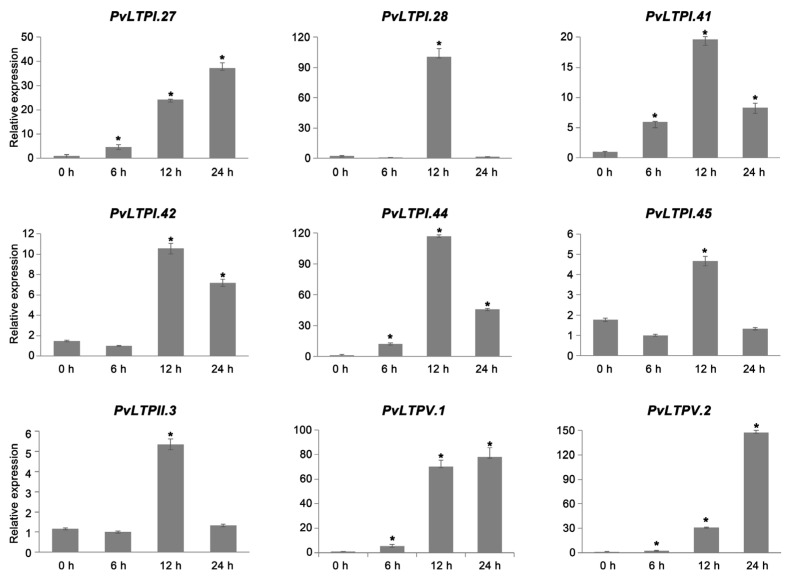
Expression levels of *PvLTPs* under drought treatment at 0, 6, 12, and 24 h by qRT-PCR. The internal reference gene used was *Actin*. Error bars reflect the standard error. A significant difference is represented by an asterisk (* *p* < 0.05).

**Table 1 genes-13-02394-t001:** Putative *PvLTP*s identified in the common bean.

Gene Name	Gene ID	Chromosome Location	Strand	CDS Length (bp)	AA ^a^	SP ^b^	MP ^c^	ECM ^d^	MW ^e^ (kD)	pI ^f^	Intron
Type I											
*PvLTPI.1*	Phvul.001G062200.1	Chr01: 7446247, 7446788	+	339	112	21	91	C-X_9_-C-X_14_-CC-X_19_-CXC-X_22_-C-X_7_-C	9956.67	6.39	1
*PvLTPI.2*	Phvul.001G062300.1	Chr01: 7454215, 7454792	+	366	121	21	100	C-X_9_-C-X_14_-CC-X_19_-CXC-X_22_-C-X_13_-C	10,991.71	6.47	1
*PvLTPI.3*	Phvul.001G062400.1	Chr01: 7469181, 7469533	+	339	112	21	91	C-X_9_-C-X_14_-CC-X_19_-CXC-X_22_-C-X_7_-C	10,048.77	6.37	1
*PvLTPI.4*	Phvul.001G062500.1	Chr01: 7497470, 7497805	+	336	111	21	90	C-X_9_-C-X_14_-CC-X_19_-CXC-X_22_-C-X_7_-C	9908.63	7.68	0
*PvLTPI.5*	Phvul.005G045100.1	Chr05: 4768172, 4769125	+	348	115	23	92	C-X_9_-C-X_14_-CC-X_19_-CXC-X_21_-C-X_13_-C	9162.50	8.49	1
*PvLTPI.6*	Phvul.005G103800.1	Chr05: 32275019, 32275716	+	378	125	22	103	C-X_9_-C-X_16_-CC-X_19_-CXC-X_23_-C-X_13_-C	11,656.84	5.56	1
*PvLTPI.7*	Phvul.006G026700.1	Chr06: 10142385, 10142738	+	354	117	26	91	C-X_9_-C-X_15_-CC-X_19_-CXC-X_21_-C-X_10_-C	9849.69	8.86	0
*PvLTPI.8*	Phvul.006G026600.1	Chr06: 10157925, 10158278	–	354	117	26	91	C-X_9_-C-X_15_-CC-X_19_-CXC-X_21_-C-X_10_-C	10,487.43	8.68	0
*PvLTPI.9*	Phvul.006G026400.1	Chr06: 10189268, 10189630	+	363	120	25	95	C-X_9_-C-X_14_-CC-X_22_-CXC-X_21_-C-X_10_-C	10,517.50	8.85	0
*PvLTPI.10*	Phvul.006G026300.1	Chr06: 10209345, 10209689	–	345	114	16	98	C-X_9_-C-X_15_-CC-X_19_-CXC-X_21_-C-X_10_-C	10,644.64	8.86	0
*PvLTPI.11*	Phvul.006G026200.1	Chr06: 10224541, 10224903	+	363	120	25	95	C-X_9_-C-X_14_-CC-X_22_-CXC-X_21_-C-X_10_-C	9762.57	8.68	0
*PvLTPI.12*	Phvul.006G026040.1	Chr06: 10279348, 10279692	–	345	114	23	91	C-X_9_-C-X_15_-CC-X_19_-CXC-X_21_-C-X_10_-C	9756.61	8.68	0
*PvLTPI.13*	Phvul.006G025800.1	Chr06: 10298843, 10299187	–	345	114	16	98	C-X_9_-C-X_15_-CC-X_19_-CXC-X_21_-C-X_10_-C	10,155.73	6.65	0
*PvLTPI.14*	Phvul.006G025840.1	Chr06: 10329063, 10329407	–	345	114	16	98	C-X_9_-C-X_15_-CC-X_19_-CXC-X_21_-C-X_10_-C	10,512.52	8.86	0
*PvLTPI.15*	Phvul.006G025880.1	Chr06: 10335897, 10336241	–	345	114	23	91	C-X_9_-C-X_15_-CC-X_19_-CXC-X_21_-C-X_10_-C	10,152.68	5.59	0
*PvLTPI.16*	Phvul.006G026800.1	Chr06: 10397866, 10398287	–	369	122	26	96	C-X_9_-C-X_14_-CC-X_19_-CXC-X_21_-C-X_10_-C	9863.40	4.52	0
*PvLTPI.17*	Phvul.006G026900.1	Chr06: 10460655, 10461008	–	354	117	26	91	C-X_9_-C-X_15_-CC-X_19_-CXC-X_21_-C-X_10_-C	10,060.47	5.04	0
*PvLTPI.18*	Phvul.006G025700.1	Chr06: 11140768, 11141112	–	345	114	16	98	C-X_9_-C-X_15_-CC-X_19_-CXC-X_21_-C-X_10_-C	10,342.94	7.02	0
*PvLTPI.19*	Phvul.006G025500.1	Chr06: 11156824, 11157168	–	345	114	23	91	C-X_9_-C-X_15_-CC-X_19_-CXC-X_21_-C-X_10_-C	9810.22	6.00	0
*PvLTPI.20*	Phvul.006G078300.1	Chr06: 19007898, 19008577	+	354	117	23	94	C-X_9_-C-X_13_-CC-X_19_-CXC-X_22_-C-X_13_-C	10,175.73	8.60	1
*PvLTPI.21*	Phvul.006G078400.1	Chr06: 19017732, 19018561	+	354	117	23	94	C-X_9_-C-X_13_-CC-X_19_-CXC-X_22_-C-X_13_-C	10,147.72	8.77	1
*PvLTPI.22*	Phvul.006G112211.1	Chr06: 22115372, 22115937	–	348	115	19	96	C-X_9_-C-X_14_-CC-X_19_-CXC-X_21_-C-X_14_-C	10,345.16	8.66	1
*PvLTPI.23*	Phvul.007G022400.1	Chr07: 1575864, 1576217	+	354	117	26	91	C-X_9_-C-X_15_-CC-X_19_-CXC-X_21_-C-X_10_-C	9900.34	5.77	0
*PvLTPI.24*	Phvul.007G067200.1	Chr07: 6183402, 6183764	–	363	120	28	92	C-X_9_-C-X_16_-CC-X_19_-CXC-X_21_-C-X_13_-C	9739.17	3.94	0
*PvLTPI.25*	Phvul.008G083400.1	Chr08: 8154052, 8155082	+	372	123	29	94	C-X_9_-C-X_15_-CC-X_19_-CXC-X_22_-C-X_13_-C	9595.25	9.07	1
*PvLTPI.26*	Phvul.008G083600.1	Chr08: 8162377, 8163555	–	369	122	28	94	C-X_9_-C-X_14_-CC-X_19_-CXC-X_23_-C-X_13_-C	10,404.81	5.09	1
*PvLTPI.27*	Phvul.008G087000.1	Chr08: 8524161, 8525277	+	351	116	25	91	C-X_9_-C-X_13_-CC-X_19_-CXC-X_21_-C-X_13_-C	9540.98	8.72	1
*PvLTPI.28*	Phvul.008G087101.1	Chr08: 8535136, 8536072	+	351	116	25	91	C-X_9_-C-X_13_-CC-X_19_-CXC-X_21_-C-X_13_-C	9444.64	4.99	1
*PvLTPI.29*	Phvul.008G197700.1	Chr08: 54196013, 54196496	–	372	123	21	102	C-X_9_-C-X_14_-CC-X_19_-CXC-X_22_-C-X_13_-C	11,251.25	9.07	1
*PvLTPI.30*	Phvul.008G197800.1	Chr08: 54201804, 54202157	–	354	117	21	96	C-X_9_-C-X_14_-CC-X_19_-CXC-X_22_-C-X_13_-C	10,563.37	8.98	0
*PvLTPI.31*	Phvul.008G197914.1	Chr08: 54214451, 54214786	–	336	111	21	90	C-X_9_-C-X_14_-CC-X_19_-CXC-X_22_-C-X_7_-C	10,021.71	8.18	0
*PvLTPI.32*	Phvul.008G198000.1	Chr08: 54231141, 54231476	–	336	111	21	90	C-X_9_-C-X_14_-CC-X_19_-CXC-X_22_-C-X_7_-C	9936.60	7.68	0
*PvLTPI.33*	Phvul.008G198200.1	Chr08: 54263601, 54264568	–	385	124	19	105	C-X_9_-C-X_14_-CC-X_19_-CXC-X_22_-C-X_7_-C	11,647.61	7.64	1
*PvLTPI.34*	Phvul.008G198300.1	Chr08: 54270808, 54271422	–	366	121	21	100	C-X_9_-C-X_14_-CC-X_19_-CXC-X_22_-C-X_13_-C	10,966.66	5.77	1
*PvLTPI.35*	Phvul.008G198414.1	Chr08: 54281566, 54282126	–	366	121	21	100	C-X_9_-C-X_14_-CC-X_19_-CXC-X_22_-C-X_13_-C	10,575.31	8.12	1
*PvLTPI.36*	Phvul.008G198400.1	Chr08: 54291418, 54291771	–	354	117	20	97	C-X_9_-C-X_14_-CC-X_19_-CXC-X_22_-C-X_13_-C	11,039.75	6.47	0
*PvLTPI.37*	Phvul.008G198500.1	Chr08: 54317230, 54317860	–	366	121	21	100	C-X_9_-C-X_14_-CC-X_19_-CXC-X_22_-C-X_13_-C	11,028.73	6.12	1
*PvLTPI.38*	Phvul.008G198700.1	Chr08: 54336612, 54337182	–	366	121	21	100	C-X_9_-C-X_14_-CC-X_19_-CXC-X_22_-C-X_13_-C	11,036.74	5.72	1
*PvLTPI.39*	Phvul.008G198900.1	Chr08: 54353165, 54353635	–	372	123	21	102	C-X_9_-C-X_14_-CC-X_19_-CXC-X_22_-C-X_13_-C	11,302.92	5.37	1
*PvLTPI.40*	Phvul.008G199100.1	Chr08: 54393051, 54393632	–	363	120	21	99	C-X_9_-C-X_14_-CC-X_19_-CXC-X_22_-C-X_13_-C	10,319.89	4.88	1
*PvLTPI.41*	Phvul.010G066500.1	Chr10: 24968137, 24969974	+	348	115	24	91	C-X_9_-C-X_13_-CC-X_19_-CXC-X_22_-C-X_13_-C	9396.85	9.22	1
*PvLTPI.42*	Phvul.010G066400.1	Chr10: 25019438, 25021034	+	357	118	25	93	C-X_9_-C-X_13_-CC-X_19_-CXC-X_23_-C-X_13_-C	9375.79	9.30	1
*PvLTPI.43*	Phvul.010G079000.2	Chr10: 25151667, 25158544	+	357	118	25	93	C-X_9_-C-X_15_-CC-X_19_-CXC-X_22_-C-X_13_-C	9221.70	9.22	1
*PvLTPI.44*	Phvul.010G069100.1	Chr10: 30206639, 30208294	+	357	118	25	93	C-X_9_-C-X_13_-CC-X_19_-CXC-X_22_-C-X_13_-C	10,412.79	8.86	1
*PvLTPI.45*	Phvul.010G069500.1	Chr10: 30837346, 30838454	–	378	125	28	97	C-X_9_-C-X_14_-CC-X_19_-CXC-X_26_-C-X_13_-C	10,144.63	10.34	1
Type II											
*PvLTPII.1*	Phvul.004G002700.1	Chr04: 126913, 127490	–	267	88	19	69	C-X_7_-C-X_13_-CC-X_8_-CXC-X_26_-C-X_7_-C	7235.32	8.73	0
*PvLTPII.2*	Phvul.004G002600.1	Chr04: 129918, 130493	–	282	93	25	68	C-X_7_-C-X_13_-CC-X_8_-CXC-X_26_-C-X_6_-C	7282.43	8.50	0
*PvLTPII.3*	Phvul.004G154200.1	Chr04: 45768730, 45769379	+	279	92	24	68	C-X_7_-C-X_13_-CC-X_8_-CXC-X_26_-C-X_6_-C	7306.61	9.33	0
*PvLTPII.4*	Phvul.007G031600.1	Chr07: 2518719, 2519872	–	285	94	18	76	C-X_7_-C-X_13_-CC-X_8_-CXC-X_26_-C-X_6_-C	7995.24	8.66	0
*PvLTPII.5*	Phvul.008G008200.1	Chr08: 734888, 735748	–	297	98	30	68	C-X_7_-C-X_13_-CC-X_8_-CXC-X_26_-C-X_6_-C	7147.38	9.03	0
Type IV											
*PvLTPIV.1*	Phvul.006G050100.1	Chr06: 15479738, 15480395	+	315	104	27	77	C-X_9_-C-X_15_-CC-X_9_-CXC-X_22_-C-X_7_-C	8316.77	8.68	1
*PvLTPIV.2*	Phvul.007G001800.1	Chr07: 109767, 110152	+	306	101	25	76	C-X_9_-C-X_15_-CC-X_9_-CXC-X_24_-C-X_7_-C	7982.32	4.34	0
*PvLTPIV.3*	Phvul.008G112900.1	Chr08: 13096130, 13097071	+	306	101	27	74	C-X_9_-C-X_15_-CC-X_9_-CXC-X_22_-C-X_7_-C	7758.24	8.90	0
*PvLTPIV.4*	Phvul.008G137100.1	Chr08: 27792568, 27793248	–	312	103	27	76	C-X_9_-C-X_15_-CC-X_9_-CXC-X_24_-C-X_7_-C	8031.21	4.22	0
*PvLTPIV.5*	Phvul.011G047400.1	Chr11: 4266384, 4267082	–	309	102	25	77	C-X_9_-C-X_15_-CC-X_9_-CXC-X_24_-C-X_7_-C	8429.80	6.70	0
Type V											
*PvLTPV.1*	Phvul.003G282100.1	Chr03: 51931056, 51932514	+	360	119	27	92	C-X_14_-C-X_14_-CC-X_12_-CXC-X_24_-C-X_10_-C	9363.27	9.90	1
*PvLTPV.2*	Phvul.008G113900.1	Chr08: 13258029, 13259010	+	351	116	24	92	C-X_15_-C-X_14_-CC-X_11_-CXC-X_24_-C-X_10_-C	9501.92	8.69	1
Type VIII											
*PvLTPVIII.1*	Phvul.002G060600.1	Chr02: 6661973, 6662884	–	465	154	42	112	C-X_6_-C-X_12_-CC-X_12_-CXC-X_25_-C-X_8_-C	12,167.10	8.88	1

^a^ The number of amino acids; ^b,c^ number of amino acids in the signal peptide and mature protein; ^d^ eight-cysteine motif; ^e,f^ molecular weight, and isoelectric point of the mature protein, respectively.

**Table 2 genes-13-02394-t002:** Diversity of EMC in five types of PvLTPs.

PvLTP Type	Spacing Pattern												
		1		2		3 4		5 6		7		8	
I	X2–3, 5–12	C	X9	C	X13–16	CC	X19, 22	CXC	X21–23, 26	C	X7, 10, 13, 14	C	X2–6, 8, 17
II	X2, 10	C	X7	C	X13	CC	X8	CXC	X23	C	X6, 7	C	X0
IV	X3, 4	C	X9	C	X15	CC	X9	CXC	X22, 24	C	X7	C	X0, 3
V	X3	C	X14, 15	C	X14	CC	X11, 12	CXC	X24	C	X10	C	X6

“X” indicates any amino acid, and the Arabic digits following “X” indicate the number of amino acids.

## Data Availability

Not applicable.

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
