# Peer review of "Genome-Wide Identification of Common Bean *PvLTP* Family Genes and Expression Profiling Analysis in Response to Drought Stress"

_genes, 2022, doi:10.3390/genes13122394_

Round 1
Reviewer 1 Report
This article studied Genome-wide Identification of Common Bean PvLTP Family Genes and Expression Profiling Analysis in Response to Drought Stress. The study is important in genetic point of view as significant findings are presented in the study specifically in stress environment. There are some shortcomings that should be resolve.
Abstract is well written but I am wonder that it follows the journal format or not.
The headings should be removed from the abstract.
Line 17 replace crops with crop.
Line 49 should be cited with more studies.
Line 52 should be cited.
Add details of the economic and industrial importance of the common bean.
Section 2.2 should be cited with relevant study.
Section 2.3 should be cited with relevant study.
section 2.4 also lack reference.
Figure 7 should be place in results section.
Provide future recommendations in the conclusion.
Reviewer 2 Report
The paper titled ‘Genome-wide identification of common Bean PV LTP family genes and expression profiling analysis in response to drought stress’ takes up an important issue with many practical applications.
Following comments are submitted
1. The paper is well-written, the writing style is comprehensive and clear.
2. The introduction is effective in in its purpose as it suitably introduces the relevance and background of the study undertaken, however, a brief introduction of various genes involved in drought stress management in Phaseolus vulgaris is missing, these should also be mentioned here.
3. Methodology is well-planned.
4. Results section has some modifications to be made, such as,
i. Line no.184-186: authors say 11 were discarded from 70 and the rest 58 were selected. The calculation doesn’t match.
ii. There are some concerns regarding expression profiling studies.
iii. Authors should give geo-tagged photographs of the seedlings grown in the growth chamber.
iv. Likewise, the geo-tagged photograph of the field should also be given as a supplementary file.
v. Authors selected 9 genes for confirming the expression, the basis of selection should be elaborated.
vi. Ct plots of all the selected genes and control gene for all the samples should also be supplied.
5. The discussion part needs improvement, particularly while discussing expression profiling. Few lines regarding upregulation and downregulation of genes under different conditions are needed to correlate to the mechanism involved.
